# Semaglutide combined with empagliflozin vs. monotherapy for non-alcoholic fatty liver disease in type 2 diabetes: Study protocol for a randomized clinical trial

Yu-Hao Lin[1,☯], Zhi-Jun Zhang[1,☯], Jin-Qing Zhong[2], Zhi-Yi Wang[1], Yi-Ting Peng[3], Yan-Mei Lin[1], Huo-Ping Zhang[4], Jian-Qing Tian[1]*

1 Department of Endocrinology, Xiamen Humanity Hospital, Fujian Medical University, Xiamen, Fujian, China, 2 Department of Laboratory Medicine, Xiamen Humanity Hospital, Fujian Medical University, Xiamen, Fujian, China, 3 Department of Endocrinology, Zhongshan Hospital Xiamen University, Xiamen, Fujian, China, 4 Department of Ultrasound, Xiamen Humanity Hospital, Fujian Medical University, Xiamen, Fujian, China

☯ These authors contributed equally to this work.
* 449256119@qq.com

**Data Availability Statement:** No datasets were generated or analysed during the current study. All

## Abstract

### Background

Nonalcoholic fatty liver disease (NAFLD) is strongly associated with type 2 diabetes mellitus (T2DM). Lifestyle intervention remains a preferred treatment modality for NAFLD. The glucagon-like peptide (GLP-1) receptor agonists and sodium-glucose cotransporter-2 (SGLT-2) inhibitors have been developed as new glucose-lowering drugs, which can improve fatty liver via an insulin-independent glucose-lowering effect. However, studies exploring the efficacy of GLP-1 receptor agonists combined with SGLT-2 inhibitors in patients with NAFLD and T2DM are scanty. Thus, the present randomised controlled trial aims at comparing the efficacy and safety of semaglutide plus empagliflozin with each treatment alone in patients with NAFLD and T2DM.

### Methods

This 52-week double-blinded, randomised, parallel-group, active-controlled trial evaluates the effects of semaglutide, empagliflozin and semaglutide + empagliflozin in 105 eligible overweight/obese subjects with NAFLD and T2DM. The primary outcome will be a change from baseline to week 52 in the controlled attenuation parameter, free fatty acid and glucagon. Secondary endpoints include changes in liver stiffness measurement, liver enzymes, blood glucose, lipid levels, renal function, electrolyte balances, minerals and bone metabolism, cytokines, high-sensitivity C-reactive protein, ferritin, anthropometric indicators, nonalcoholic fatty liver fibrosis score, fibrosis 4 score and homeostatic model assessment for insulin resistance. In addition, intention-to-treat, interim analysis and safety analysis will be performed.

relevant data from this study will be made available upon study completion.

**Funding:** This work was supported by the Medical and Health Guidance Projects of Xiamen, China (Grant No.3502Z20224ZD1106). The sponsors or funders not involved in the study design, data collection and analysis, decision to publish, or preparation of the manuscript.

## Discussion

This double-blinded, randomised, clinical trial involves a multi-disciplinary approach and aims to explore the synergistic effects of the combination of semaglutide and empagliflozin. The results can provide important insights into mechanisms of GLP-1 receptor agonists and/or SGLT-2 inhibitors in patients with NAFLD and T2DM.

## Trial registration

This study has been registered with Chinese Clinical Trial Registry (ChiCTR2300070674).

## Introduction

Nonalcoholic fatty liver disease (NAFLD) is characterized by the presence of hepatic steatosis exceeding 5% in individuals without excessive alcohol intake or any other chronic liver diseases [1]. The prevalence of NAFLD has increased steadily by 0.7% annually, from 21.9% to 37.3% between 1990 and 2019 [2]. This rise in prevalence is associated with a growing elderly population and a rise in the incidence of type 2 diabetes mellitus (T2DM) and obesity [3]. Over 55% of T2DM patients complicated with NAFLD and suboptimal glycemic control is a prominent clinical risk factor for more advanced forms of NAFLD, including fibrosis, cirrhosis, hepatocellular carcinoma and liver failure [4,5].

T2DM and NAFLD share pathophysiological features that act synergistically to increase the risk of metabolic syndrome through dysregulation of glucose and lipid metabolism as well as insulin resistance [6]. Increased adiposity and insulin resistance contribute to the stimulation of *de novo* lipogenesis, resulting in elevated levels of free fatty acids [7]. Consequently, there is an accumulation of ectopic fat in liver cells, pancreas, or muscles, which ultimately results in the formation of fatty liver and pancreatic steatosis [8]. Hence, there is a high unmet need among patients with T2DM and NAFLD for efficacious pharmacotherapies to improve liver health, provide glycemic control and mitigate the risk of cardiovascular and renal morbidity. Unfortunately, no pharmacotherapies have yet been approved by the United States Food and Drug Administration (FDA) and the European Medicines Agency (EMA) for the treatment of NAFLD. Newer glucose-lowering agents, including glucagon-like peptide (GLP-1) receptor agonists and sodium-glucose cotransporter-2 (SGLT-2) inhibitors have been assessed in participants with T2DM and NAFLD, because of their multiple pleiotropic effects [9,10].

To our knowledge, semaglutide, a GLP-1 receptor agonist, has been associated with an improvement in liver-enzyme levels and amelioration of the severity of hepatic steatosis [11–13]. Further, a real-world investigation has demonstrated that semaglutide can improve liver steatosis in patients with T2DM [14]. Similarly, empagliflozin, an SGLT-2 inhibitor, has been shown to improve liver steatosis in T2DM patients with NAFLD or nonalcoholic steatohepatitis (NASH) [15–18]. Concomitant treatment with GLP-1 receptor agonists and SGLT-2 inhibitors may have synergistic effects due to their distinct mechanisms. Compared with monotherapy, the combination of exenatide, a GLP-1 receptor agonist, and dapagliflozin, an SGLT-2 inhibitor, showed superior efficacy in ameliorating markers of hepatic steatosis and fibrosis in patients with T2DM [19]. Currently, there is a dearth of studies that may guide the concomitant use of GLP-1 receptor agonists and SGLT-2 inhibitors, and no existing proof regarding the use of semaglutide and empagliflozin combination in treating patients with T2DM and NAFLD [20].

Hence, the present prospective randomised controlled trial (RCT) sought to investigate the efficacy and safety of semaglutide plus empagliflozin compared with each treatment alone in patients with T2DM and NAFLD.

# Methods

## Trial design and setting

A schematic diagram of the study design is displayed in Fig 1. This is a double-blinded, randomised, parallel-group, active-controlled trial of patients with T2DM and NAFLD. A total 105

| | STUDY PERIOD | | | | | | |
|---|---|---|---|---|---|---|---|
| | Enrolment | Allocation | Post-allocation | | | | Close-out |
| TIMEPOINT* | | | Baseline / Week 0 | Week 12 | Week 24 | Week 52 | $t_x$ |
| **ENROLMENT:** | | | | | | | |
| Eligibility screen | X | | | | | | |
| Informed consent | X | | | | | | |
| Randomization | | X | | | | | |
| Allocation | | X | | | | | |
| **INTERVENTIONS:** | | | | | | | |
| semaglutide | | | ←———————————→ | | | | |
| empagliflozin | | | ←———————————→ | | | | |
| semaglutide plus empagliflozin | | | ←———————————→ | | | | |
| **ASSESSMENTS:** | | | | | | | |
| Demographic and clinical characteristics | X | X | X | | | | |
| **Primary outcome** | | | | | | | |
| CAP | X | | X | | X | X | X |
| FFA | | | X | X | X | X | |
| glucagon | | | X | X | X | X | |
| **Secondary outcomes** | | | | | | | |
| LSM | X | | X | | X | X | X |
| liver enzymes | | | X | X | X | X | X |
| fasting glucose | X | | X | X | X | X | X |
| HbA1c | | | X | X | X | X | X |
| fasting insulin | | | X | X | X | X | |
| fasting C-peptide | | | X | X | X | X | |
| lipid profiles | | | X | X | X | X | |
| renal function and electrolyte balances | | | X | X | X | X | |
| Minerals and bone metabolism | | | X | | X | X | |
| cytokines | | | X | X | X | X | |
| hsCRP | | | X | X | X | X | |
| ferritin | | | X | X | X | X | |
| anthropometric indicators | X | | X | X | X | X | |
| NFS | | | X | X | X | X | |
| FIB-4 | | | X | X | X | X | |
| HOMA-IR | | | X | X | X | X | |
| **Safety outcomes** | | | | | | | |
| Adverse events | | | X | X | X | X | X |

**Fig 1. Schedule of enrolment, interventions, and assessments.** Notes: CAP, controlled attenuation parameter; FFA, free fatty acid; LSM, liver stiffness measurement; NFS, Nonalcoholic fatty liver fibrosis score; FIB-4, Fibrosis 4 score; HOMA-IR, Homeostatic model assessment for insulin resistance.

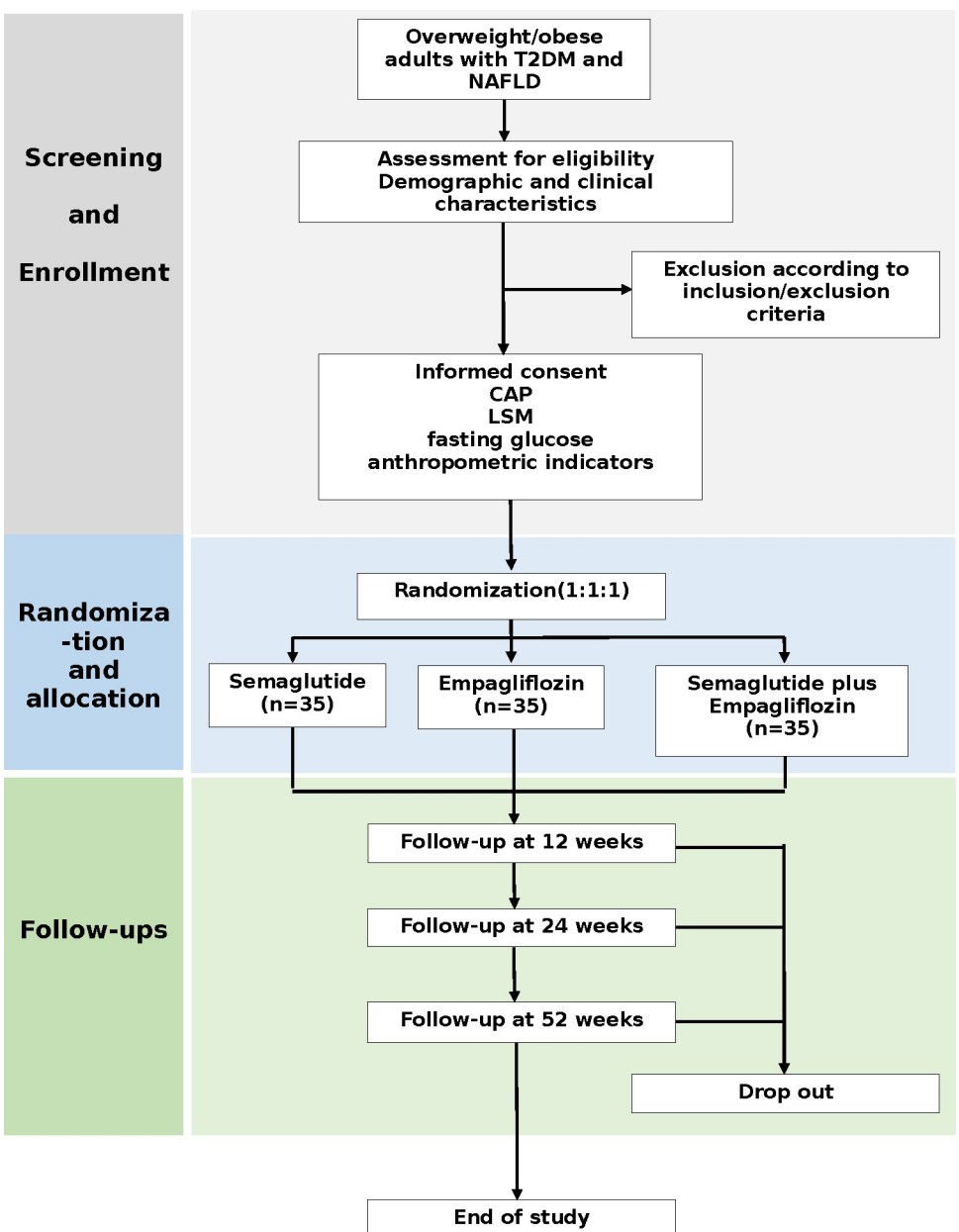

**Fig 2. Flow of participants.** Notes: T2DM, type 2 diabetes mellitus; NAFLD, nonalcoholic fatty liver disease; CAP, controlled attenuation parameter; LSM, liver stiffness measurement.

participants divided in a 1:1:1 ratio will be recruited, of which 35 will be randomised to semaglutide, 35 to empagliflozin and 35 to semaglutide plus empagliflozin. All patients will be recruited from Xiamen Humanity Hospital, of Fujian Medical University. Included patients will receive the trial drug and be monitored for 52 weeks. The trial will be completed in 2025. All clinical visits will be conducted at Xiamen Humanity Hospital of Fujian Medical University. All investigators will undergo good clinical practice training and be trained in the study requirements, standardized clinical measurements and counselling for adherence. A flow chart of the study process is shown in Fig 2.

Written informed consent will be obtained from all patients who will voluntarily agree to participate in this study. The study design and present protocol follows the guidelines of the Standard Protocol Items: Recommendations for Interventional Trials (SPIRIT) and the Declaration of Helsinki [21].

## Patient and public involvement

Patients and/or the public were not involved in the design, or conduct, or reporting, or dissemination plans of this research.

## Eligibility criteria

Volunteers expressing interest will undergo eligibility screening based on the inclusion and exclusion criteria.

## Inclusion criteria

1. Age of $\geq$18 years old at the time of enrolment.

2. Diagnosis of NAFLD includes: evidence of excess liver fat accumulation (hepatic steatosis) on ultrasound imaging and no history of alcohol overconsumption and exclusion of other specific causes of fatty liver [22].

3. Diagnosis of T2DM includes: glycated hemoglobin (HbA1c) $\geq$ 6.5% (48 mmol/mol) or fasting plasma glucose $\geq$ 126 mg/dl (7.0 mmol/L) or 2-h plasma glucose $\geq$ 200 mg/dl (11.1 mmol/L) during an oral glucose tolerance test or in an individual with classic symptoms of hyperglycemia or hyperglycemic crisis, a random plasma glucose $\geq$ 200 mg/dL (11.1 mmol/L). In the absence of unequivocal hyperglycemia, diagnosis requires two abnormal test results obtained at the same time or at two different time points [23].

4. Body mass index (BMI) more than 24 kg/m$^2$.

5. HbA1c $\geq$ 6.5% and $\leq$ 10.5%.

6. Liver enzyme values less than three times the upper limit of normal.

7. Patients who had not used GLP-1 receptor agonists and SGLT-2 inhibitors within 4 weeks.

## Exclusion criteria

1. Type 1 diabetes mellitus, gestational diabetes mellitus or other types of diabetes mellitus.

2. Previously treated with thiazolidinediones or insulin.

3. History of cardiac, hepatic or renal insufficiency.

4. Patients with viral hepatitis (such as hepatitis B), alcoholic liver disease, autoimmune hepatitis, drug-induced liver disease, hemochromatosis, Wilson's disease, liver cirrhosis, inborn errors of metabolism (such as cholesterol ester storage disease), or other causes of chronic liver disease.

5. History of cerebral stroke, malignant tumor or pancreatitis.

6. History of medullary thyroid carcinoma or multiple endocrine neoplasia type 2 in oneself or family.

7. Pregnancy, lactation or desire for conception during the study period.

8. Consumed more than 140g of ethanol per week for men and more than 70g of ethanol per week for women.

9. Known or suspected hypersensitivity to GLP-1 receptor agonists, SGLT-2 inhibitors or metformin.

10. Known or suspected mental and psychological disorders.

11. History of chronic anaemia (Hb hemoglobin level <100 g/L in men and <90 g/L in women).

12. Unwilling or unable to provide informed consent.

## Interventions

Following the completion of all evaluations, participants will be randomly assigned to three groups. Each group will receive either semaglutide plus placebo, empagliflozin plus placebo or semaglutide plus empagliflozin. The subcutaneous semaglutide injection will be initiated at a dosage of 0.25 mg once weekly for the first 4 weeks and the dose will be doubled every 4 weeks until 1 mg is reached (days 1–28: 0.25 mg semaglutide, days 29–56: 0.5 mg semaglutide and days 57–364: 1.0 mg semaglutide). Patients who cannot tolerate uptitration to 1.0 mg semaglutide will continue with 0.5 mg once weekly. Patients who cannot tolerate 0.5 mg/0.25 mg semaglutide plus placebo/empagliflozin plus placebo/semaglutide plus empagliflozin will be excluded from the study. Patients will receive instruction on the subcutaneous injection technique, which can be self-administered at home. Patients who cannot self-inject will be assisted by a nurse will. Patients will receive 10 mg of empagliflozin daily throughout the study in the empagliflozin plus placebo or semaglutide plus empagliflozin group. Patients will be required to report any development of adverse events (AEs) during the entire treatment period to their provider. Besides taking study medications, patients will be advised on the management of various coexisting illnesses during the trial. Before the commencement of the trial, all participants will receive nutritional and exercise counseling from an experienced dietician. In between follow-up interviews, participants will receive a reminder of scheduled study visits via WeChat.

## Outcomes

After obtaining the participants' informed consent, demographic and clinical characteristics of the participants will be collected, including age, sex, height, weight, waist circumference, medical history and medications, course of disease and family health history of T2DM, etc.

## Primary outcome

Since this is an exploratory trial, primary outcomes will include: (1) controlled attenuation parameter (CAP); (2) free fatty acid; and (3) glucagon.

## Secondary outcomes

Secondary endpoints include changes in (1) liver stiffness measurement (LSM), (2) liver enzymes, (3) fasting glucose, HbA1c, fasting insulin and fasting C-peptide, (4) lipid profiles, (5) renal function and electrolyte balances, (6) minerals and bone metabolism (25-hydroxyvitamin D3 (25[OH]D3) and parathyroid hormone (PTH)), (7) cytokines (interleukin-6 (IL-6) and adiponectin), (8) high-sensitivity C-reactive protein (hsCRP), (9) ferritin, (10)

anthropometric indicators (body weight, BMI, the waist and hip circumference, waist-to-hip ratio, numbers of weight loss achieving at least 5% and blood pressure), (11) nonalcoholic fatty liver fibrosis score (NFS) [24]: $1.675 + 0.037 \times$ age (years) $+ 0.094 \times$ BMI (kg/m$^2$) $+ 1.13 \times$ impaired fasting glycaemia (IFG)/diabetes (yes = 1, no = 0) + (0.99 $\times$ aspartate aminotransferase (AST)/alanine aminotransferase (ALT) ratio) (0.013 $\times$ platelet [$\times$109/L]) (0.66 $\times$ albumin [g/dl]), (12) Fibrosis 4 score (FIB-4) [25]: (age [years] $\times$ AST [U/L])/([platelets (109/L)] $\times \sqrt{}$ ALT [U/L]), (13) homeostatic model assessment for insulin resistance (HOMA-IR) [26]: [fasting glucose (mmol/l) $\times$ fasting insulin (pmol/l)]/135.

## Safety outcomes

Participants will be required to report any AEs, and details, including time of occurrence, severity, duration, treatment and outcome, will be recorded. AEs will be continuously monitored. In addition, all reported AEs will be explained and resolved properly.

## Outcome measurements

The FibroScan is equipped with both M and XL probes, and a probe selection that depended on real-time assessment of the skin-to-liver capsule distance for each participant [27]. All participants will be required to fast for at least 2 hours before the FibroScan test. Assessment of hepatic steatosis and fibrosis in the participants will be evaluated with CAP and LSM, respectively using FibroScan at baseline and, weeks 24 and 52. After 12 hours of fasting, 10 mL of blood sample will be taken from each participant by a certified phlebotomist. The sample will be centrifuged at 3500 rpm for 10 min for serum extraction. The level of free fatty acid, glucagon, liver enzymes (ALT, AST and gamma-glutamyl transferase (GGT)), fasting glucose, HbA1c, fasting insulin, fasting C-peptide, serum lipid profiles (total cholesterol (TC), triglyceride (TG), low-density lipoprotein-cholesterol (LDL-C) and high-density lipoprotein-cholesterol (HDL-C)), renal function, electrolyte balances, 25[OH]D3, PTH, IL-6, adiponectin, hsCRP and ferritin will be measured by an accredited medical laboratory. These measurements will be conducted at baseline and weeks 12, 24 and 52. Outcomes will be assessed by independent investigators, who will be blinded to the grouping assignment. The results of the grouping will be assigned a number and then placed inside a sealed envelope.

## Sample size

The Professional Association for SQL Server (PASS) 2021 software will be used for sample size calculation. This calculation will be based on the estimate of the CAP as reported by previous RCT study [28]. Considering a two-tailed test with α = 0.05 and 90% power, the total number of participants will be 93, with 31 participants in each group. Assuming a dropout rate of 10%, the total number of participants will be 105, with 35 participants in each group.

## Recruitment

Participants with T2DM and NAFLD will be recruited from outpatient and inpatient of the endocrinology or gastroenterology department of the Xiamen Humanity Hospital of Fujian Medical University. Notices about recruitment are displayed on bulletin boards within the hospital, as well as on WeChat friend circle, WeChat group, Weibo and other social media platforms. The notices will encompass a concise overview of the study aims, medicines, eligibility criteria, contact information and information on how to participate in the study. The eligibility of potential participants will be evaluated by the study coordinator. Thereafter, the study coordinator will screen eligible participants based on inclusion and exclusion criteria

and provide a detailed explanation of the study protocol to ensure that participants fully understand the trial. All participants will be required to sign an informed consent form if they agree to participate in the study. Recruitment started in April 14, 2023 and will continue until the end of June 30, 2024.

## Randomization

Participants who enrolled in the study and provided their written informed consent will complete the baseline assessment. Afterwards, participants will be randomly assigned to each of the matched groups in a 1:1:1 allocation ratio using permutation blocks with a block size of 4, stratified according to age, gender and BMI. A randomisation list will be generated by an independent researcher using the Sealed Envelope online randomization program. The randomisation sequence and block sizes will be concealed in a sealed opaque envelope until after enrollment. The allocation assignment will be performed by an independent researcher who will not be involved in the outcome assessment. Similarly, participants, healthcare providers, data analysts and outcome assessors will be blinded to the treatment allocation.

## Data collection and management

Data entry, coding, security, and storage will be performed by a well-trained study coordinator throughout the trial. Data will be collected at baseline, and weeks 12, 24 and 52. All participants will first undergo a point-of-care FibroScan to assess the degree of fatty liver, followed by blood sample collection to assess metabolic parameters, renal function, electrolyte balances, inflammatory cytokines, minerals and bone metabolism, as well as biometric assessments to evaluate demographic information. Clinical research coordinators will collect all required data from participants' self-administered questionnaire. Clinical data and viral test results of the participants will be collected in a standardised electronic case report forms and transferred to a uniform electronic data capture (EDC) system. To ensure confidentiality, all data will be anonymised before being submitted to other project staff. The database will be locked until completion of the study. All data will be archived at Xiamen Humanity Hospital for at least 5 years or longer after completion of the study.

## Statistical analysis

The Shapiro-Wilk test will be used to evaluate the normal distribution of continuous variables. Normally distributed data will be represented as mean ± standard deviation (SD), while non-normally distributed data will be expressed as medians and interquartile ranges (IQRs). Data for categorical variables will be summarized using numbers, percentages and frequencies. Mixed-effect model for repeated measures (MMRM) will be applied to examine the effect of treatment on longitudinal assessments of continuous efficacy end-points, with the corresponding end point and baseline value in the model. Analysis of covariance (ANCOVA) will be used for within-group comparisons among three treatment groups with age and sex as factors, and baseline body weight and baseline biomarker as covariates. For categorical efficacy endpoints, logistical regression with fixed effects of treatment and baseline as a covariate will be used. Safety analyses of AEs in both groups and the comparison among three groups will be conducted using the Chi-squared test or Fisher's exact test. Additionally, statistical analyses will be performed using a modified intention-to-treat (mITT) principle, which includes all patients who received at least one dose of study drug and had at least one postbaseline measurement of any outcome. Missing data for change from baseline in outcomes was estimated by longitudinal integrated two-component prediction model. Statistical analysis will be performed using R statistical software V.4.3.1. Statistical significance will be set as two-sided p-values of $\leq 0.05$.

An interim analysis is planned for this study when a 50% enrollment is reached. Patient information and clinical data will remain confidential on a professional data platform.

## Trial monitoring

**Data monitoring.**   A Data Monitoring Committee (DMC) has been established, encompassing one endocrinologist, one hepatologist and one biostatistician who are familiar with clinical trial design. The DMC will regularly review the study progress and advise on study continuation and/or any changes to the protocol. Results from interim analysis will be reported to the DMC and blinded to the investigators. Further, the DMC is independent of the trial sponsor and has no conflicting interests.

**Harms.**   Information about AEs will be collected on a questionnaire throughout the intervention period plus two additional weeks and recorded in case report forms with details such as occurrence time, duration, severity, treatment, outcome, and relevance to the treatment. AEs will be assessed by the study team via out-patient hospital visits, phone calls or WeChat. Overall, semaglutide and empagliflozin is relatively safe and may be associated with gastrointestinal adverse effects and urinary infections. If this occurs, consider taking symptomatic treatment. AEs are defined as serious if they result in adverse reactions requiring intensive treatment or fatality. Any AEs and serious AEs will be immediately reported to the DMC and the principal investigator. Afterwards, the principal investigator will immediately take appropriate measures in response to serious AEs. For other significant AEs, a case-by-case discussion will be conducted.

**Auditing.**   Data auditing will be conducted by an independent auditor trimonthly throughout the trial and near the time that is planned for interim analyses. Specifically, 10% of the questionnaires will be randomly drawn and inspected regarding their degree of matching with the database input.

## Ethics and dissemination

**Research ethics approval.**   This clinical trial has been approved by the Research Ethics Committee of Xiamen Humanity Hospital of Fujian Medical University (HAXM-MEC-20230105-002-01). All trial associated procedures will be conducted to comply with the Declaration of Helsinki principles.

**Protocol amendments.**   Protocol amendments will be carried out according to Good Clinical Practice requirements and approved by the Research Ethics Committee. Any protocol amendments will be documented and explained.

**Consent or assent.**   A trained investigator will be responsible for obtaining informed assent and consent from potential participants before participating in the study. Informed consent will be provided to participants in written and verbal. If requested, verbal translation will also be provided. The consent process will include providing the participants with a detailed introduction to the study and fully answering any questions raised by the participants.

**Confidentiality.**   Personal information for both potential and enrolled participants will be kept confidential before, during and after the trial. Standardised paper forms will be used to collect personal information from participants. To ensure confidentiality, each enrolled participant will be assigned a personal identification number, which will also be used for subsequent data analysis. Participant's personal information will be stored on a secure password-protected server and will not be included in any study forms, reports, publications or any other disclosures unless legally mandated.

Declaration of interests

No competing interests or conflicts of interest to be disclosed.

Access to data

Access to the final trial dataset will be restricted to the data managers and study investigators. However, to ensure confidentiality, any identifying participant information will be blinded from project team members.

Ancillary and post-trial care

No ancillary or post-trial care is planned.

Dissemination policy

The results from the research will be submitted for publication in a peer-reviewed journal and will also be presented at relevant scientific conferences with personal information omitted. And the trial results will be communicated collectively to the participants, who will then be given the option to request a copy of the final report or a simplified version thereof, which will be provided upon their preference. As part of our media outreach efforts, the findings will be shared with healthcare professionals and the public via our webpages, WeChat, and Microblog. Eligibility for authorship will be determined following the guidelines set by the International Committee of Medical Journal Editors (ICMJE).

## Discussion

NAFLD is a major public health concern with an increasing prevalence worldwide. The coexistence of this condition with diabetes poses a considerable threat, exacerbating the likelihood of cirrhosis, liver cancer and mortality [29]. Since there are currently no effective therapies for T2DM with NAFLD yet, the identification of safe and efficacious pharmacological treatments poses a significant challenge. Therefore, we will conduct a 52-week double-blinded randomised trial to compare the efficacy of semaglutide plus empagliflozin versus semaglutide or empagliflozin monotherapy.

Nowadays, histopathological evaluation of liver biopsy is regarded as the gold standard for diagnosing NAFLD, determining disease stage and predicting prognosis [30]. However, liver biopsy presents several limitations, such as the invasive nature of sampling, patient reluctance, and postoperative risks, especially when repeated sampling is required. Given the limitations of repeated liver biopsy, ultrasound will be used to observe changes in CAP and LSM among patients with T2DM and NAFLD in our study. In addition, patients' free fatty acid, glucagon, relevant indexes for glucose and lipid metabolism, insulin sensitivity, noninvasive scores, adiponectin and other indicators will be evaluated.

GLP-1 receptor agonists and SGLT-2 inhibitors exert beneficial effects on hepatic lipotoxicity and inflammation through mechanisms that may, in part, be independent of weight reduction [16,31]. However, the dominant underlying mechanism by which GLP-1 receptor agonists and SGLT-2 inhibitors reduce liver fat remains elusive. Recently, Pedersen et al. found that NAFLD impair the liver–alpha cell axis and cause glucagon resistance [32]. Hepatic glucagon resistance towards amino acid catabolism and lipid metabolism follows. This results in increased plasma levels of amino acids and free fatty acids, contributing to further secretion of glucagon [33]. We hypothesize that GLP-1 receptor agonists and SGLT-2 inhibitors improve glucagon resistance, thereby increasing β-oxidation and decreasing lipogenesis, resulting in a reduction of hepatic triglyceride content [34]. It has been recently suggested that SGLT-2 inhibitors downregulate miR-34a-5p expression and inhibit the expression of genes related to the transforming growth factor beta (TGFβ) signaling pathway in hepatic stellate cells to ameliorate NAFLD-associated fibrosis [35]. Our study seeks to explore the synergistic effects of GLP-1 receptor agonist and SGLT2 inhibitor combination.

## Supporting information

**S1 Checklist. Reporting checklist for protocol of a clinical trial.**
(DOCX)

**S1 File.**
(DOCX)

**S2 File.**
(DOCX)

## Acknowledgments

We would like to thank Medical and Health Guidance Projects of Xiamen, China for funding this work.

## Author Contributions

**Conceptualization:** Yu-Hao Lin, Zhi-Jun Zhang.

**Data curation:** Jin-Qing Zhong.

**Formal analysis:** Yu-Hao Lin, Huo-Ping Zhang.

**Funding acquisition:** Jian-Qing Tian.

**Investigation:** Zhi-Jun Zhang, Zhi-Yi Wang, Yi-Ting Peng.

**Methodology:** Yan-Mei Lin, Huo-Ping Zhang.

**Project administration:** Zhi-Jun Zhang, Jin-Qing Zhong, Zhi-Yi Wang, Yan-Mei Lin.

**Resources:** Jin-Qing Zhong, Huo-Ping Zhang.

**Software:** Yu-Hao Lin.

**Supervision:** Zhi-Yi Wang, Yi-Ting Peng, Jian-Qing Tian.

**Validation:** Yi-Ting Peng, Jian-Qing Tian.

**Visualization:** Yan-Mei Lin.

**Writing – original draft:** Yu-Hao Lin.

**Writing – review & editing:** Zhi-Jun Zhang, Jian-Qing Tian.

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
