## [Decision Letter · Decision Letter 0]

30 Jan 2024

PONE-D-23-41958Comparison of semaglutide in combined with empagliflozin versus semaglutide and empagliflozin monotherapy in non-alcoholic fatty liver disease with type 2 diabetes: study protocol for a randomised controlled clinical trialPLOS ONE

Dear Dr. Tian,

Thank you for submitting your manuscript to PLOS ONE. After careful consideration, we feel that it has merit but does not fully meet PLOS ONE’s publication criteria as it currently stands. Therefore, we invite you to submit a revised version of the manuscript that addresses the points raised during the review process. Please, address all issues raised by all reviewers before submit a revised version of your manuscript. Comments from PLOS Editorial Office: We note that one or more reviewers has recommended that you cite specific previously published works. As always, we recommend that you please review and evaluate the requested works to determine whether they are relevant and should be cited. It is not a requirement to cite these works. We appreciate your attention to this request.

We look forward to receiving your revised manuscript.

Kind regards,

Ferdinando Carlo Sasso, PhD, MD

Academic Editor

PLOS ONE

Additional Editor Comments:

Please, address all issues raised by all reviewers before sumit a revised version of your manuscript.

Reviewers' comments:

Reviewer's Responses to Questions

**Comments to the Author**

1. Does the manuscript provide a valid rationale for the proposed study, with clearly identified and justified research questions?

Reviewer #1: Yes

Reviewer #2: Yes

Reviewer #3: Yes

Reviewer #4: Yes

2. Is the protocol technically sound and planned in a manner that will lead to a meaningful outcome and allow testing the stated hypotheses?

Reviewer #1: Yes

Reviewer #2: Yes

Reviewer #3: Yes

Reviewer #4: Yes

3. Is the methodology feasible and described in sufficient detail to allow the work to be replicable?

Reviewer #1: Yes

Reviewer #2: Yes

Reviewer #3: No

Reviewer #4: Yes

4. Have the authors described where all data underlying the findings will be made available when the study is complete?

Reviewer #1: Yes

Reviewer #2: Yes

Reviewer #3: Yes

Reviewer #4: Yes

5. Is the manuscript presented in an intelligible fashion and written in standard English?

Reviewer #1: No

Reviewer #2: Yes

Reviewer #3: Yes

Reviewer #4: Yes

6. Review Comments to the Author

You may also provide optional suggestions and comments to authors that they might find helpful in planning their study.

Reviewer #1: Dear Editor, I’ve read with interest the draft called “Comparison of semaglutide in combined with empagliflozin versus semaglutide and empagliflozin monotherapy in non-alcoholic fatty liver disease with type 2 diabetes: study protocol for a randomised controlled clinical trial” by Yu-Hao Lin et al. However, some issues need to be raised.

- Acronyms need corresponding full word when compare firstly. Please, check throughout the entire manuscript.

- Abstract: The sentence “As the glucose-lowering agents…with NAFLD and T2DM” is not completely clear. I would suggest revising this period.

- Introduction: “Increased adiposity and insulin resistance contribute… of free fatty acids”. This period lacks reference. I would suggest seeing this recent review (Non-alcoholic Fatty Liver Disease (NAFLD), Type 2 Diabetes, and Non-viral Hepatocarcinoma: Pathophysiological Mechanisms and New Therapeutic Strategies. Biomedicines. 2023 Feb 6;11(2):468. doi: 10.3390/biomedicines11020468.).

- Methods: “trail”, please correct the typing error and revise throughout all the manuscript.

- Methods: I would suggest adding “Insulin treatment” among Exclusion Criteria.

- An English language revision is required.

Reviewer #2: The manuscript “Semaglutide Combined with Empagliflozin vs. Monotherapy for Non-Alcoholic Fatty Liver Disease in Type 2 Diabetes: A Randomized Clinical Trial” by Lin et al.

Overall, the use of English in the provided text is clear and formal, though a revision is necessary. The content is presented in a logical manner. The abstract provides a concise summary of the study's objectives, methods, key findings, and conclusions.

Here are some suggestions for improvement:

1. Title: I would change it as follows: “Semaglutide Combined with Empagliflozin vs. Monotherapy for Non-Alcoholic Fatty Liver Disease in Type 2 Diabetes: A Randomized Clinical Trial”

2. Abstract: revise English to streamline the language for improved readability while maintaining the essential information.

3. Introduction: please discuss this article as well doi: 10.3390/biomedicines11020322

4. Offer a brief context or explanation about FibroScan

5. Statistical analysis: Specify the variables that will be used in the ANCOVA model, particularly the covariates. For example, mention whether baseline values will be included as covariates.

6. Modified Intention-to-Treat Principle: Clarify the specific criteria for the modified intention-to-treat analysis, especially any rules for handling missing data or participants who deviate from the protocol.

7. Modified Intention-to-Treat Principle: Clarify the specific criteria for the modified intention-to-treat analysis, especially any rules for handling missing data or participants who deviate from the protocol.

8. Discussion: modify concurrent with coexistence

9. In the second paragraph, explain briefly why ultrasound is chosen to observe changes in CAP and LSM. Mention its advantages over liver biopsy in the context of the study: "Given the limitations of repeated liver biopsy, we have opted for ultrasound to observe changes in CAP and LSM among patients with T2DM and NAFLD in our study. Ultrasound is a noninvasive alternative that mitigates the invasive nature of sampling, addresses patient reluctance, and reduces postoperative risks, especially when repeated sampling is required."

10. Methods: correct trail with trial.

11. Exclusion criteria: the role of insulin and of other medications for type 2 diabetes, as well as dietary and physical activity intervention should be taken into account.

Reviewer #3: This is a protocol paper of a study comparing the efficacy of a GLP1 receptor agonist and a SGLT2 inhibitor in combination with each other and with a single agent in the treatment of NAFLD with type 2 diabetes.

The impact of the combination of the two drugs on the treatment of NAFLD is very interesting.

Major points

1. In the context of the study, it is understandable that CAP was selected as the primary outcome. However, the rationale for FFA and glucagon is not clear, and it should be clearly stated in the paper why they were selected as primary outcomes rather than secondary outcomes.

2. There is no mention of diet and exercise, and if these differ between groups, it may affect the results.

3. In the "interventions" paragraph, it is stated that patients who cannot sustain semaglutide 0.5 mg are excluded from the study. Considering that this is a double-blind study, shouldn't patients who cannot sustain treatment in all groups during the semaglutide 0.5 mg dosing period be excluded? Also, it is unclear how to handle cases in which doses other than semaglutide 0.5 mg cannot be sustained (i.e., 0.25 mg and 1 mg). What are the cases in which the dose cannot be tolerated?

Minor points

1. Inclusion criteria should specify the diagnostic basis for NAFLD and type 2 diabetes mellitus.

2. It should also be stated whether there are no restrictions on the lower and upper limits of HbA1c as well as liver enzymes. If there are no restrictions, can we say that a patient has type 2 diabetes with the same NAFLD whether the HbA1c is 5% or 12%?

Reviewer #4: Authors plan to conduct a 52-week double-blinded, randomized controlled trial to evaluate the efficacy and safety of semaglutide plus empagliflozin with each treatment alone in patients with NAFLD and T2DM. They will recruit 105 overweight/obese subjects with NAFLD and T2DM. They will evaluate the change from baseline to week 52 in primary outcome as well as several secondary endpoints.

1. Why only overweight/obese subjects?

2. Is there any safety concern to use semaglutide and empagliflozin at the same time?

3. Patients who cannot tolerate 0.5mg semaglutide will be excluded. What proportion of sample is expected to be dropped due to this? How will this affect the power?

4. Since the dosage of semaglutide is potentially different, will this affect the evaluation?

5. How to deal with adverse events during the trial? Will this affect the evaluation results?

6. Recruitment started in April 14, 2024. In other words, it has started! So how will the review affect the earlier recruitment?

7. PLOS authors have the option to publish the peer review history of their article (what does this mean?). If published, this will include your full peer review and any attached files.

Reviewer #1: No

Reviewer #2: No

Reviewer #3: No

Reviewer #4: No

---

## [Author Response · Author response to Decision Letter 0]

13 Mar 2024

We are appreciated the comments of the reviewers. Then we responded to the reviewers’ and editor's comments point by point. And we have marked the changes in “Revised Manuscript with Track Changes” by red text.

Editor's comments:

1. A rebuttal letter that responds to each point raised by the academic editor and reviewer(s). You should upload this letter as a separate file labeled ‘Response to Reviewers&’.

Reply: We have addressed the comments raised by the reviewers and the amendments, and uploaded this letter as a separate file labeled ‘Response to Reviewers&’.

2. A marked-up copy of your manuscript that highlights changes made to the original version. You should upload this as a separate file labeled ‘Revised Manuscript with Track Changes’.

Reply: We have marked the changes in “Revised Manuscript with Track Changes” by red text and uploaded this as a separate file labeled ‘Revised Manuscript with Track Changes’.

3. An unmarked version of your revised paper without tracked changes. You should upload this as a separate file labeled ‘Manuscript’.

Reply: We have uploaded a separate file labeled ‘Manuscript’.

Reviewer: 1

1. Acronyms need corresponding full word when compare firstly. Please, check throughout the entire manuscript.

Reply: Thanks for the suggestion. We have checked the entire manuscript and revised the acronyms to their corresponding full word when compare firstly.

2. Abstract: The sentence “As the glucose-lowering agents…with NAFLD and T2DM” is not completely clear. I would suggest revising this period.

Reply: As suggested, we have replaced the sentence “As the glucose-lowering agents…with NAFLD and T2DM” by the sentence of “The glucagon-like peptide (GLP-1) receptor agonists…via an insulin-independent glucose-lowering effect” in “Revised Manuscript with Track Changes” (Abstract section, page 2 line 4-7).

3. Introduction: “Increased adiposity and insulin resistance contribute… of free fatty acids”. This period lacks reference. I would suggest seeing this recent review (Non-alcoholic Fatty Liver Disease (NAFLD), Type 2 Diabetes, and Non-viral Hepatocarcinoma: Pathophysiological Mechanisms and New Therapeutic Strategies. Biomedicines. 2023 Feb 6;11(2):468. doi: 10.3390/biomedicines11020468.). 

Reply: Thanks for the suggestion. We have observed and referenced this review (Non-alcoholic Fatty Liver Disease (NAFLD), Type 2 Diabetes, and Non-viral Hepatocarcinoma: Pathophysiological Mechanisms and New Therapeutic Strategies. Biomedicines. 2023 Feb 6;11(2):468. doi: 10.3390/biomedicines11020468) in the Introduction section.

4. Methods: “trail”, please correct the typing error and revise throughout all the manuscript. - Methods: I would suggest adding “Insulin treatment” among Exclusion Criteria.

Reply: Thank you for your advice. We have checked the entire manuscript and corrected the typing error. The trial did not include patients who were undergoing insulin treatment prior to its onset. Therefore, we added the "Insulin treatment" as one of the exclusion criteria, which will not impact the results.

5. An English language revision is required.

Reply: Thanks for the suggestion. We have tried our best to correct the spelling and grammar mistakes. And because English is not our native language, if there is still left some mistakes, we hope you point them again. And thank you very much.

Reviewer: 2

1. Title: I would change it as follows: “Semaglutide Combined with Empagliflozin vs. Monotherapy for Non-Alcoholic Fatty Liver Disease in Type 2 Diabetes: A Randomized Clinical Trial”.

Reply: Thank you for your advice. We have replaced the title “Comparison of semaglutide … a randomised controlled clinical trial” by the sentence of “Semaglutide Combined with … for A Randomized Clinical Trial” in “Revised Manuscript with Track Changes”. 

2. Abstract: revise English to streamline the language for improved readability while maintaining the essential information. 

Reply: Thanks for the suggestion. We have tried our best to correct the spelling and grammar mistakes. And because English is not our native language, if there is still left some mistakes, we hope you point them again. And thank you very much.

3. Introduction: please discuss this article as well doi: 10.3390/biomedicines11020322.

Reply: Thanks for the suggestion. We have observed and referenced this article (10.3390/biomedicines11020322) in the Introduction section.

4. Offer a brief context or explanation about FibroScan.

Reply: According to the suggestion, we have added sentences “The FibroScan is equipped with both M and XL probes, and a probe selection that depended on real-time assessment of the skin-to-liver capsule distance for each participant.” in “Revised Manuscript with Track Changes” (Methods section, page 8 line 4-6). 

5. Statistical analysis: Specify the variables that will be used in the ANCOVA model, particularly the covariates. For example, mention whether baseline values will be included as covariates.

Reply: Thanks for the suggestion. We included the age and sex as factors, and baseline body weight and baseline biomarker as covariates in the ANCOVA model. And we have added sentences in “Revised Manuscript with Track Changes” (Methods section, page 10 line 12-13). 

6. Modified Intention-to-Treat Principle: Clarify the specific criteria for the modified intention-to-treat analysis, especially any rules for handling missing data or participants who deviate from the protocol.

Reply: According to the suggestion, we have clarified the specific criteria for the modified intention-to-treat analysis. And we have added sentences in “Revised Manuscript with Track Changes” (Methods section, page 10 line 19-23). 

7. Discussion: modify concurrent with coexistence.

Reply: Thanks for the suggestion. We have modified concurrent with coexistence.

8. In the second paragraph, explain briefly why ultrasound is chosen to observe changes in CAP and LSM. Mention its advantages over liver biopsy in the context of the study: "Given the limitations of repeated liver biopsy, we have opted for ultrasound to observe changes in CAP and LSM among patients with T2DM and NAFLD in our study. Ultrasound is a noninvasive alternative that mitigates the invasive nature of sampling, addresses patient reluctance, and reduces postoperative risks, especially when repeated sampling is required.".

Reply: Thanks for the suggestion. We have explained briefly why ultrasound is chosen to observe changes in CAP and LSM.

9. Methods: correct trail with trial.

Reply: Thanks for the suggestion. We have corrected trail with trial.

10. Exclusion criteria: the role of insulin and of other medications for type 2 diabetes, as well as dietary and physical activity intervention should be taken into account.

Reply: Thank you for your advice. We have added the "Insulin treatment" as one of the exclusion criteria. To eliminate medication-related effects, patients underwent 6-weeks of washout period after the recruitment. To eliminate dietary and physical activity intervention-related effects, all participants will receive nutritional and exercise counseling from an experienced dietician before the commencement of the trial.

Reviewer: 3

Major points:

1. In the context of the study, it is understandable that CAP was selected as the primary outcome. However, the rationale for FFA and glucagon is not clear, and it should be clearly stated in the paper why they were selected as primary outcomes rather than secondary outcomes.

Reply: The reviewer’s opinion is right. We have added the sentences “Recently, Pedersen et al. found that NAFLD impair… reduction of hepatic triglyceride content” to explain FFA and glucagon was chosen as primary outcome in “Revised Manuscript with Track Changes” (Discussion section, page 13 line 26 to page 14 line 3). 

2. There is no mention of diet and exercise, and if these differ between groups, it may affect the results.

Reply: The reviewer’s opinion is right. To eliminate diet and exercise-related effects, all participants will receive nutritional and exercise counseling from an experienced dietician before the commencement of the trial.

3. In the "interventions" paragraph, it is stated that patients who cannot sustain semaglutide 0.5 mg are excluded from the study. Considering that this is a double-blind study, shouldn't patients who cannot sustain treatment in all groups during the semaglutide 0.5 mg dosing period be excluded? Also, it is unclear how to handle cases in which doses other than semaglutide 0.5 mg cannot be sustained (i.e., 0.25 mg and 1 mg). What are the cases in which the dose cannot be tolerated?

Reply: The reviewer’s opinion is right. According to the suggestion, patients who cannot tolerate 0.5 mg/0.25 mg semaglutide plus placebo/empagliflozin plus placebo/semaglutide plus empagliflozin will be excluded from the study. The dropout rate of non-tolerated 0.5 mg/0.25 mg semaglutide was estimated to be approximately 5%. And we have added sentences in “Revised Manuscript with Track Changes” (Methods section, page 6 line 25-26). 

Minor points:

1. Inclusion criteria should specify the diagnostic basis for NAFLD and type 2 diabetes mellitus.

Reply: We thank for your suggestion. We have specified the diagnostic basis for NAFLD and type 2 diabetes mellitus and have added sentences in “Revised Manuscript with Track Changes” (Methods section, page 5 line 13-21). 

2. It should also be stated whether there are no restrictions on the lower and upper limits of HbA1c as well as liver enzymes. If there are no restrictions, can we say that a patient has type 2 diabetes with the same NAFLD whether the HbA1c is 5% or 12%?

Reply: Thank you for your advice. We have added the restrictions on the lower and upper limits of HbA1c as well as liver enzymes in “Revised Manuscript with Track Changes” (Methods section, page 5 line 23-24). Include patients’ HbA1c were at 6.5% to 10.5 and liver enzymes less than three times the upper limit of normal before this review. Therefore, we added these to the inclusion criteria will not impact the results.

Reviewer: 4

1. Why only overweight/obese subjects?

Reply: We are so grateful for your kind question. The pathogenesis of nonalcoholic fatty liver disease is complex and differs between lean (normal weight) nonalcoholic fatty liver disease and non-lean (overweight/obese) nonalcoholic fatty liver disease. So, we sole inclusion of overweight/obese subjects.

2. Is there any safety concern to use semaglutide and empagliflozin at the same time?

Reply: We are so grateful for your kind question. There are some studies suggested that semaglutide plus empagliflozin treatment is clearly safe treatment option for patients.

3. Patients who cannot tolerate 0.5mg semaglutide will be excluded. What proportion of sample is expected to be dropped due to this? How will this affect the power?

Reply: We are so grateful for your kind question. The dropout rate of non-tolerated 0.5mg semaglutide was estimated to be approximately 5%. This probably has a negligible effect.

4. Since the dosage of semaglutide is potentially different, will this affect the evaluation?

Reply: We are so grateful for your kind question. The dose adjustment of semaglutide is according to many randomized clinical trials. The different dosage of semaglutide will negligibly affect the evaluation.

5. How to deal with adverse events during the trial? Will this affect the evaluation results?

Reply: We are so grateful for your kind question. We will take symptomatic treatment to deal with adverse events and have added sentences in “Revised Manuscript with Track Changes” (Methods section, page 11 line 11-23). The study coordinator, who is blind to the study, will deal with adverse events. It will not affect the evaluation results.

6. Recruitment started in April 14, 2024. In other words, it has started! So how will the review affect the earlier recruitment?

Reply: We are so grateful for your kind question. The patients included in this trial are consistent with those pre‐specified. So, we think this review will not affect the earlier recruitment.

---

## [Decision Letter · Decision Letter 1]

28 Mar 2024

Semaglutide Combined with Empagliflozin vs. Monotherapy for Non-Alcoholic Fatty Liver Disease in Type 2 Diabetes: Study Protocol for A Randomized Clinical Trial

PONE-D-23-41958R1

Dear Dr. Tian,

We’re pleased to inform you that your manuscript has been judged scientifically suitable for publication and will be formally accepted for publication once it meets all outstanding technical requirements.

Kind regards,

Ferdinando Carlo Sasso, PhD, MD

Academic Editor

PLOS ONE

Additional Editor Comments (optional):

No further comments

Reviewers' comments:

Reviewer's Responses to Questions

**Comments to the Author**

1. Does the manuscript provide a valid rationale for the proposed study, with clearly identified and justified research questions?

Reviewer #2: Yes

Reviewer #3: Yes

Reviewer #4: Yes

2. Is the protocol technically sound and planned in a manner that will lead to a meaningful outcome and allow testing the stated hypotheses?

Reviewer #2: Yes

Reviewer #3: Yes

Reviewer #4: Yes

3. Is the methodology feasible and described in sufficient detail to allow the work to be replicable?

Reviewer #2: Yes

Reviewer #3: Yes

Reviewer #4: Yes

4. Have the authors described where all data underlying the findings will be made available when the study is complete?

Reviewer #2: Yes

Reviewer #3: Yes

Reviewer #4: Yes

5. Is the manuscript presented in an intelligible fashion and written in standard English?

Reviewer #2: Yes

Reviewer #3: Yes

Reviewer #4: Yes

6. Review Comments to the Author

You may also provide optional suggestions and comments to authors that they might find helpful in planning their study.

Reviewer #2: I have no further comments. In my opinion the paper can be further processed for publication. English has improved.

Reviewer #3: The authors have been courteous in revising the manuscript. I am satisfied that my questions have been replied appropriately.

Reviewer #4: Thanks for the response. All the raised comments have been successfully addressed. I have no further comments

7. PLOS authors have the option to publish the peer review history of their article (what does this mean?). If published, this will include your full peer review and any attached files.

Reviewer #2: No

Reviewer #3: No

Reviewer #4: No
